# Fabrication of Alumina-Doped Optical Fiber Preforms by an MCVD-Metal Chelate Doping Method

**K. A. Mat Sharif [1], N. Y. M. Omar [2] , M. I. Zulkifli [1], S. Z. Muhamad Yassin [1] and H. A. Abdul-Rashid [2],\***

[1] Telekom Malaysia Research & Development Sdn. Bhd., Cyberjaya 63000, Malaysia; khairulmcvd@gmail.com (K.A.M.S.); mohdimran@tmrnd.com.my (M.I.Z.); shahrinzen@tmrnd.com.my (S.Z.M.Y.)

[2] Fiber Optics Research Center, Faculty of Engineering, Multimedia University, Cyberjaya 63100, Malaysia; nasr-omar@hotmail.com

**\*** Correspondence: hairul@mmu.edu.my

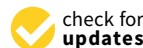

**Featured Application: Fabrication of rare-earth-doped fiber amplifiers and high-power fiber lasers.**

**Abstract:** This paper reports on the fabrication of alumina-doped preforms using a modified chemical vapor deposition (MCVD)-vapor phase chelate delivery system with $Al(acac)_3$ as the precursor. The objectives of this work are to study the deposition process, the efficiency of the fabrication process, and the quality of the fabricated fiber preforms. Two parameters are studied, the $Al(acac)_3$ sublimator temperature ($T_{Al}$ °C) and the deposition direction (i.e., downstream and upstream). Other parameters such as the oxygen flow and deposition temperature are fixed. The results show that high uniformity of the refractive index difference (%RSD < 2%) and core size (%RSD < 2.4%) was obtained along the preform length using downstream deposition, while for the combined upstream and downstream deposition, the uniformity deteriorated. The process efficiency was found to be about 21% for $T_{Al}$ °C of 185 °C and downstream deposition. From the EDX elemental analysis, the refractive index was found to increase by 0.0025 per mole percent of alumina.

**Keywords:** MCVD (Modified Chemical Vapor Deposition); chelate delivery system; $Al(acac)_3$; alumina; vapor phase doping; EDX (X-ray spectroscopy)

## 1. Introduction

Alumina ($Al_2O_3$) is an interesting dopant material for silica optical fiber technology. It does not only function as a refractive index raiser but also helps dissolve other co-dopants such as rare-earth elements in the silica matrix. Thus, alumina is very essential to the fabrication of rare-earth-doped fiber amplifiers and high-power fiber lasers. The use of alumina was first reported by Maurer and Schultz [1] in 1972 for passive fiber applications, followed by Simpson and Macchesney [2], Ohmori et al. [3], Roba [4], Wang et al. [5,6], and Čampelj et al. [7]. The fabrication techniques used were outside vapor deposition (OVD), vertical axial deposition (VAD), and modified chemical vapor deposition (MCVD). With the rapid progress on rare-earth specialty fibers, more research is being focused on incorporating alumina in silica, particularly by the MCVD method. The most common technique for doping alumina is conventional MCVD with solution doping (non-vapor phase) (e.g., [8]). This technique has been very successful in the fabrication of fiber amplifiers such as erbium-doped fiber (EDF). However, the technique suffers from limitations, including the inability to deliver advanced fiber design requirements such as high dopant concentration, large core, and precise wave guiding structure.

Furthermore, due to the nature of the MCVD solution doping process that involves several stages, it tends to degrade the quality of the preform or fiber.

Another alternative technique is MCVD with a chelate delivery system, which was first reported by Tumminelli et al. [9]. This technique offers in situ vapor phase deposition of dopants in a controllable process, thus improving the quality of the fabricated preform or fiber. Since the study by Tumminelli et al., several research groups have reported on the use of MCVD with chelate delivery systems, e.g., [10–17]. In most of these studies, anhydrous aluminum chloride ($AlCl_3$) has been used as the precursor for alumina, with few using aluminum acetylacetonate ($Al(C_5H_7O_2)_3$; $Al(acac)_3$) as a precursor, e.g., [16]. $AlCl_3$ is commonly used due to its relatively high vapor pressure. In this paper, we report on the fabrication of alumina-doped preforms using $Al(acac)_3$. Even though $Al(acac)_3$ has lower vapor pressure than $AlCl_3$, it exhibits some advantages over $AlCl_3$, including its low cost, non-corrosive properties, and chemical stability. However, the conversion of $Al(acac)_3$ to $Al_2O_3$ requires more oxygen (compared to $AlCl_3$) in order to eliminate incomplete oxidation of $Al(acac)_3$ which can lead to carbon contamination. The total gas flow rate (oxygen and carrier gas) $Q_T$ is thus considerably higher compared to that in the standard MCVD process. This entails the study of the chemistry and the deposition mechanism involved, as well as the effect of various process parameters on the deposition and incorporation efficiency of the $Al_2O_3$ particles formed during the process. In this work, we studied the fabrication process of alumina-doped silica fiber preforms with different $Al(acac)_3$ sublimator temperature ($T_{Al}$ °C) and deposition direction (i.e., downstream and upstream). Other parameters such as total oxygen flow, deposition temperature, carriage speed ($V_b$), and spindle rotation were fixed. The aim is to achieve the highest possible doping amount of alumina, which, in turn, allows for greater incorporation of rare-earth elements into silica fibers. The fabricated preforms were checked for radial and longitudinal uniformity. The efficiency of the fabrication process was also determined by comparing the $Al_2O_3$ concentration obtained from energy-dispersive X-ray spectroscopy (EDX) analyses with that derived theoretically. We are hopeful that the outcome of this study will add to the growing research concerning the MCVD-vapor phase fabrication of (highly) rare-earth-doped silica fibers for high-power fiber lasers and fiber amplifiers.

## 2. Materials and Methods

Our chelate delivery system is depicted in Figure 1. The $Al(acac)_3$ is placed in a sublimator (maximum operating temperature of 220 °C), and the vapor is carried to the reaction zone by a constant flow of high-purity helium gas using heated stainless steel tubes. A ceramic heater is placed at the end of the delivery tube and just before the MCVD main oxyhydrogen burner in order to prevent any condensation of $Al(acac)_3$ vapor on the walls of the glass substrate tube.

The fabrication process is divided into two major consecutive steps; first is the MCVD process, and second is the $Al(acac)_3$ oxidation process. These steps are then followed by a standard MCVD procedure including sintering and collapsing of the glass substrate tube to a solid preform. At the MCVD step, a Heraeus F300 synthetic silica substrate tube (25mm (Outer Diameter), 19 mm (Inner Diameter)) is first rinsed with isopropanol/acetone to remove any organic contaminants. The tube is then mounted onto a glass working lathe and is etched with $SF_6$ for several passes of the oxyhydrogen burner at a very high temperature. Several layers of high-purity $SiO_2$ are then deposited and sintered to act as a barrier between the glass substrate tube and the preform's core. Finally, for the preform's core, two layers of unsintered $SiO_2$ are deposited for doping with alumina. For the $Al(acac)_3$ oxidation step, $Al(acac)_3$ vapor is delivered to the glass substrate tube where it is converted to $Al_2O_3$ at high temperature and in the presence of high-purity oxygen gas. The deposition of the formed $Al_2O_3$ particles occurs on the surface of the deposited unsintered silica layers. Table 1 lists the process parameters used in this work. For Preform 1 (P1), $Al(acac)_3$ was sublimed at $T_{Al}$ °C of 175 °C, with carrier gas (He) and $O_2$ flow rates of 1440 and 2400 sccm, respectively. The deposition temperature ($T_{dep}$ °C) was fixed at 1850 °C with a carriage speed ($V_{dep}$) of 100 mm/min and spindle rotation of 50 rpm (rotation per minute). A total of 10 layers were deposited in the downstream direction. For Preforms 2 (P2) and 3 (P3), $T_{Al}$ °C

was fixed at 185 °C, with P2 having a total of 8 layers deposited in the downstream direction and P3 having 7 layers deposited in each direction (i.e., downstream and upstream). Figure 2 illustrates the downstream and upstream deposition directions.

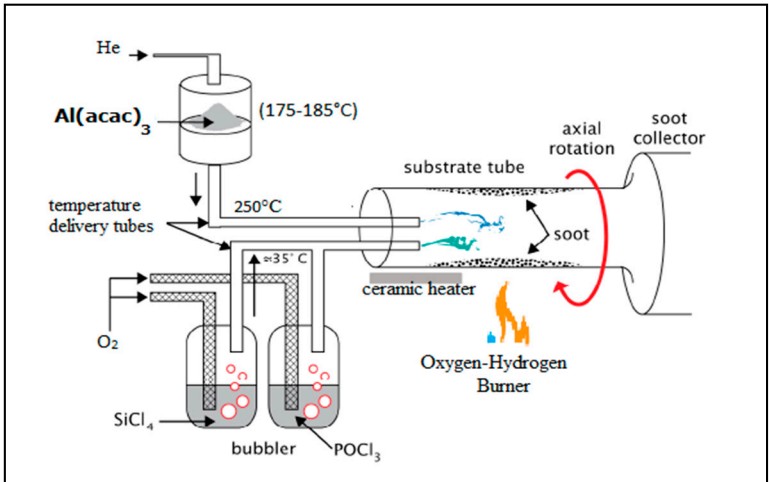

**Figure 1.** A schematic diagram for the modified chemical vapor deposition (MCVD)-chelate delivery system [14]. The Al(acac)$_3$ is placed in a sublimator consisting of a stainless steel plate, a heater, and a gas system (outlet and inlet). The stainless steel delivery tube is equipped with a heater to prevent any Al(acac)$_3$ vapor condensation. The standard MCVD delivery lines and the chelate delivery line are joined with a special rotary seal. A ceramic heater is placed just before the hot zone.

**Table 1.** Process parameters and calculated flow rates for the fabrication process.

| Preform | $T_{Al}$ (°C) | Gas Flow (sccm) | | $Q_v$ (g/min) [1] | $Q_m$ (g/min) [1] | Number of Passes | | Total $Q_m$ (g) [1] |
|---|---|---|---|---|---|---|---|---|
| | | He | O$_2$ | | | Forward | Backward | |
| P1 | 175 | | | 0.062 | $0.97 \times 10^{-2}$ | 10 | - | 0.34 |
| P2 | | 1440 | 2400 | | | 8 | - | 0.58 |
| P3 | 185 | | | 0.116 | $1.82 \times 10^{-2}$ | 7 | 7 | 1.02 |

[1] $V_{dep}$ 100 mm/min, $Q_v$ is the flow rate of reactant (g/min), $Q_m$ is the product's flow rate (g/min), and total $Q_m$ is the total amount of product during the process (g). The vapor pressure for Al(acac)$_3$ was taken from [18].

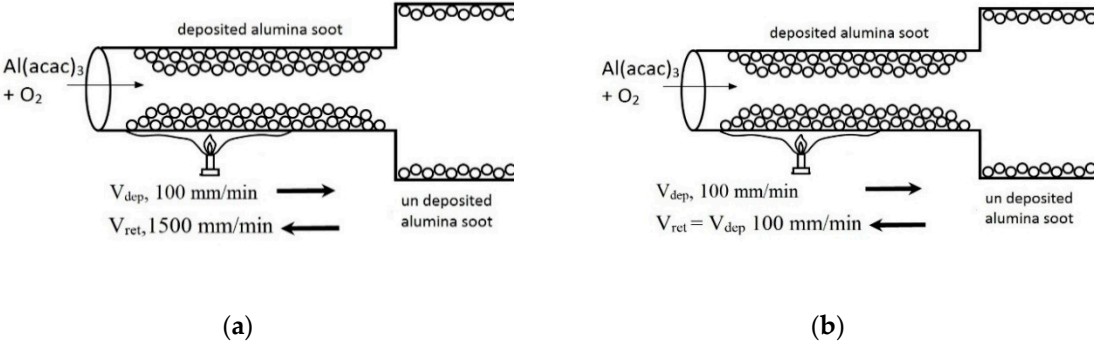

(**a**)                (**b**)

**Figure 2.** (**a**) The downstream deposition direction with $V_{dep}$ 100 mm/min and $V_{ret}$ (burner return speed) 1500 mm/min; and (**b**) downstream and upstream deposition directions ($V_{dep}$ 100 mm/min and $V_{ret} = V_{dep} = 100$ mm/min).

The refractive index profiles (RIPs) and longitudinal uniformity of the fabricated preforms were determined using a preform analyzer (*Photon Kinetics*, P104). The longitudinal uniformity of the fabricated preforms was determined by scanning the preform at 2.0 cm intervals. For each preform,

an approximately 3.0-mm-thick disk (with known index difference) was cut from the preform, polished, and subjected to EDX analyses. The EDX analyses were performed using the point identification technique with $10 \times 10$ grid points mapped in each preform's core area and an acquisition time of 240 s per point. The fabricated preforms were pulled into fibers (125 µm in diameter) using a standard fiber drawing tower and were also subjected to EDX point identification analyses in order to determine the amount and radial distribution of alumina in the core region. The efficiency of the fabrication process was determined by dividing the alumina concentration obtained from EDX analyses by that derived theoretically.

## 3. Results and Discussion

Figure 3 shows the oxidation process of Al(acac)$_3$ at T$_{dep}$ 1850 °C and an oxygen flow rate of 2400 sccm. As the Al(acac)$_3$ vapor passes through the hot zone, alumina particles (soot) are formed and then deposited on the substrate tube wall downstream of the oxyhydrogen burner due to thermophoretic forces. It was observed that the alumina soot was deposited further from the oxyhydrogen burner (i.e., longer taper region). This can be attributed to the high total gas flow rate ($Q_T$) used during the oxidation process (3840 sccm). The length over which the deposition takes place is proportional to $Q_T/\alpha$ [19], where $\alpha$ is the thermal diffusivity. Since the hot zone created by our oxyhydrogen burner is small (~2 cm in length), and $Q_T$ is high, the residence time of Al(acac)$_3$ inside the hot zone is short. This results in the formation of predominantly fine alumina particles that can be uniformly distributed along the substrate tube. Continuous movement of the hot zone results in fusion of the deposited alumina particles and incorporation into the silica soot. The deposited alumina/silica layers are then vitrified (in oxygen atmosphere) into transparent glassy material.

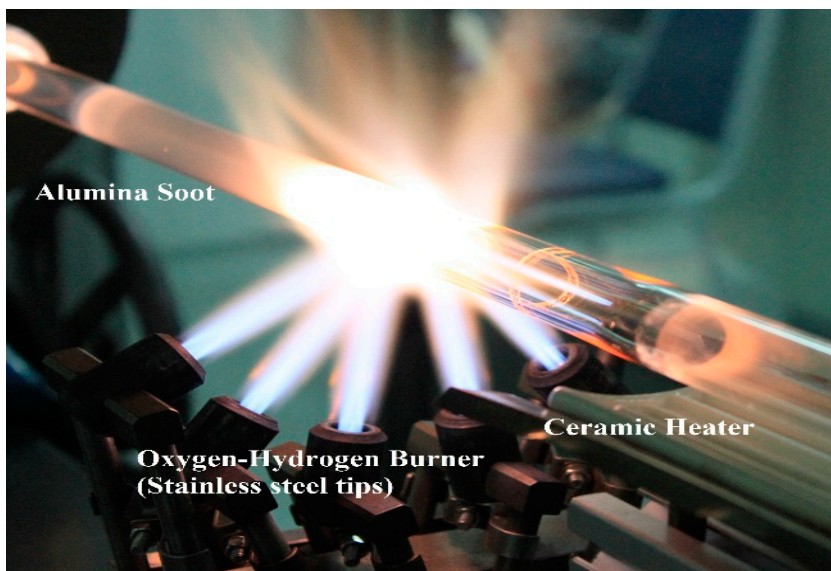

**Figure 3.** The Al(acac)$_3$ oxidation process via MCVD, where fine white particles (or alumina soot) are deposited along the glass substrate tube while the Al(acac)$_3$ vapor passes through the hot zone. The figure also shows the ceramic heater used to prevent the condensation of Al(acac)$_3$ vapor prior to reaching the hot zone.

Figure 4a,b shows the refractive index profiles for P1 (T$_{Al}$ 175 °C) and P2 (T$_{Al}$ 185 °C), respectively. Figure 4c,d displays the longitudinal uniformity values of P1 (average Δn = 0.0036) and P2 (average Δn = 0.0124), respectively. As can be seen from the figure, P1 and P2 showed good longitudinal uniformity with a slight variation in the refractive index difference (percent relative standard deviation of index difference (%RSD) 1% and 2%, respectively). This good longitudinal uniformity can be ascribed to the high total gas flow rate ($Q_T$) used and the short hot zone and

residence time, which result in the production of fine and uniform alumina particles, as discussed above. As is illustrated in Figure 5a–d, the core ($d_{core}$) and preform ($d_{preform}$) diameters for P1 were 1.2 mm (%RSD 2.4%) and 15.0 mm (%RSD 1.2%), respectively, whereas those for P2 were 1.44 mm (%RSD 1.4%) and 15.4 mm (%RSD 1.0%), respectively. This shows that it is possible to fabricate alumina-doped silica preforms with high uniformity of $\Delta n$ and core size using Al(acac)$_3$ and a chelate vapor delivery system.

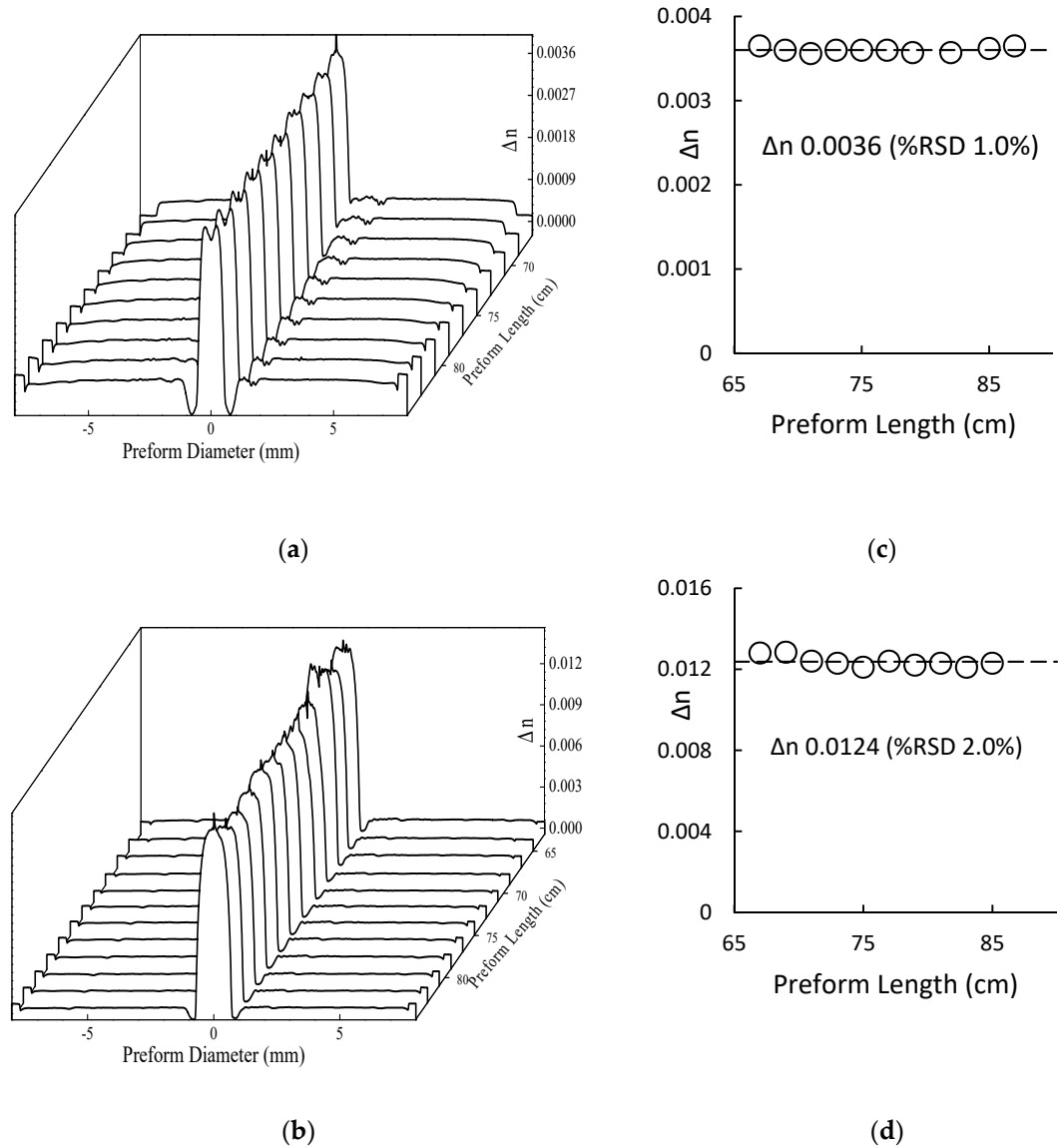

**Figure 4.** (**a**,**b**) show the refractive index profiles for P1 and P2 along the preform length; (**c**,**d**) illustrate the longitudinal variation in the refractive index difference ($\Delta n$) for P1 and P2, respectively.

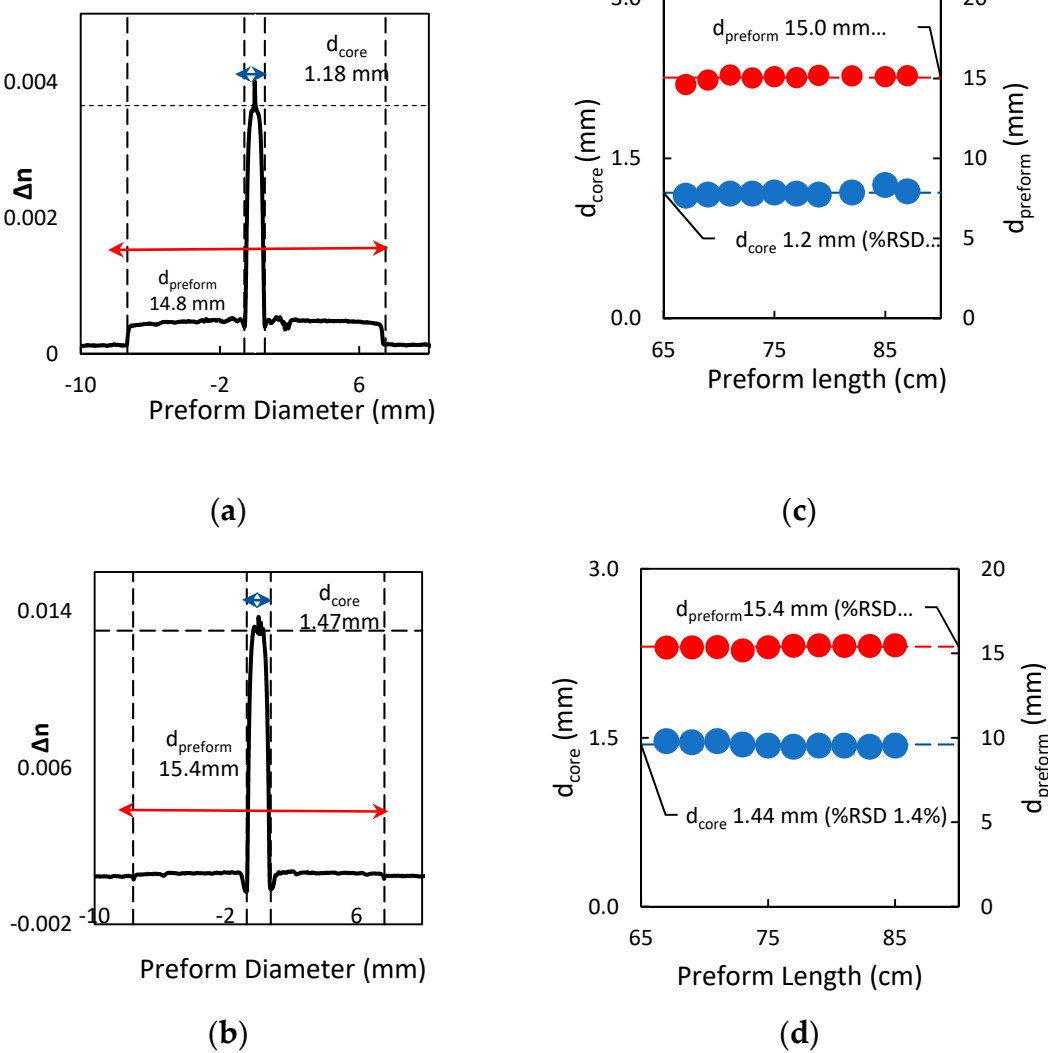

**Figure 5.** (**a**,**b**) present the refractive index profiles for P1 and P2, respectively, obtained at the inlet position and showing the measurements for $d_{preform}$ and $d_{core}$; (**c**,**d**) present the longitudinal uniformity of $d_{preform}$ and $d_{core}$ for P1 and P2, respectively.

For preform P3, the deposition of alumina was performed in both the downstream and upstream directions. The upstream deposition mode is normally used to utilize a higher temperature without sintering the produced soot [20,21]. In P3, it was observed that the downstream deposition showed the same behavior as P2. In upstream deposition, however, more alumina with large particle size distribution is produced. In this case, the alumina particles are deposited behind the moving burner and are partially sintered since the burner is moving away towards the reactant inlet. In addition, the upstream deposition mode provides higher temperature and a longer hot zone, which, in turn, result in longer residence time and enhanced conversion or oxidation of Al(acac)$_3$. This causes more particle nucleation and agglomeration and, hence, larger particle size distribution in the produced alumina soot [20]. In both P2 and P3, the effect of $Q_T/\alpha$ was the same where the length of deposition was observed to be long (i.e., taper region). It is worth mentioning that during the sintering process for P3, a red glow with intensity increasing towards the exhaust tube was observed along the substrate tube (Figure 6a). This is indicative of the high concentration of alumina and was manifest in the collapsed preform where the core had an opaque center stretching from about the middle of the preform to the outlet with increasing opacity towards the outlet (Figure 6b). This may be attributed to the deposition of Al$_2$O$_3$ in the tiny spaces between silica soot particles. The high temperatures encountered during sintering and collapse may then be enough to cause some aluminum diffusion

into the silica, thereby producing regions of alumina-rich silicates. The slow cooling of the produced materials as the burner moves away from these regions promotes solidification to a crystalline (rather than an amorphous) phase, causing opacity of the core. Another explanation is the formation of alumina-rich silicates by phase separation and crystallization. This may take place when the binary oxide mixture ($Al_2O_3$/$SiO_2$) encounters a suitable temperature during sintering or collapse, provided that the aluminum concentration is high enough [22].

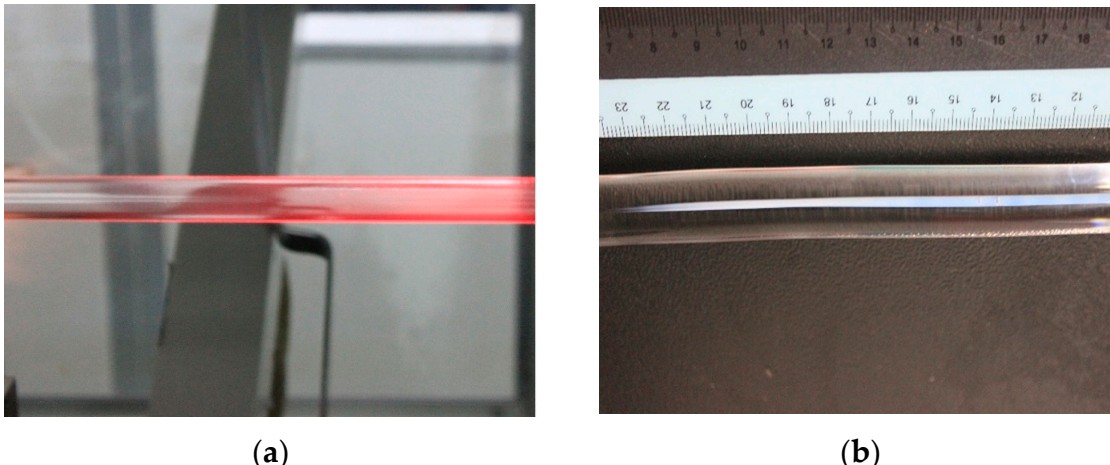

(**a**)　　　　　　　　　　　　　　　　　　　　　　(**b**)

**Figure 6.** (**a**) P3 sintering process with an oxygen flow rate of 1000 sccm, temperature of 2000–2100 °C, and $V_b$ of 125 mm/min. The red glow is an indicator of the high alumina content in the preform. (**b**) shows the opaque regions in the core due to the presence of Al-rich aluminosilicate crystallites.

It should be pointed out that even though the upstream deposition yielded more alumina, the uniformity in the longitudinal refractive index difference significantly deteriorated. However, the uniformity could be improved with a cooling mechanism, as reported by Bubnov et al. [23]. Figure 7a illustrates the refractive index profile for P3. The variation in the longitudinal refractive index difference is rather small along the first 13 cm from the inlet (%RSD 4%, Figure 7b). It is, however, significantly high along the full length of the preform. The highest core-to-cladding refractive index difference was found to be 0.027 and was for the region of the preform adjacent to the outlet. It was noticed that the opacity in the core was pronounced when the core-to-cladding refractive index difference was greater than 0.015 (~6 mol.% $Al_2O_3$). This is consistent with the conclusion that the opaque regions in the core of the preform are attributable to the formation of Al-rich aluminosilicate crystalline phase. One possible aluminosilicate crystalline phase is mullite ($3Al_2O_3.2SiO_2$). X-ray diffraction and Raman spectroscopy analyses carried out by Abramov et al. [24] indicated that high-temperature annealing of aluminosilicate fibers and preforms gives rise to the formation of crystalline mullite phase. It should be mentioned that a rapid cooling rate may prevent crystallization in the core of the preform, although such a rate is typically much higher than that achieved during the MCVD process.

The energy-dispersive X-ray spectroscopy (EDX) method was used to investigate the Al distribution and content across the core of the preforms and fibers. The EDX point identification analyses were carried out to support and complement the afore-discussed refractive index profile results. The EDX results for preform P3 are illustrated in Figure 8a,b. As can be seen from Figure 8a, the Al distribution across the core region is uniform and matches the RIP where the Al content gradually increases, reaches a maximum, and then gradually decreases. The Al concentration is highest in the center of the core and was found to range from 8.6 to 9.9 wt.%.

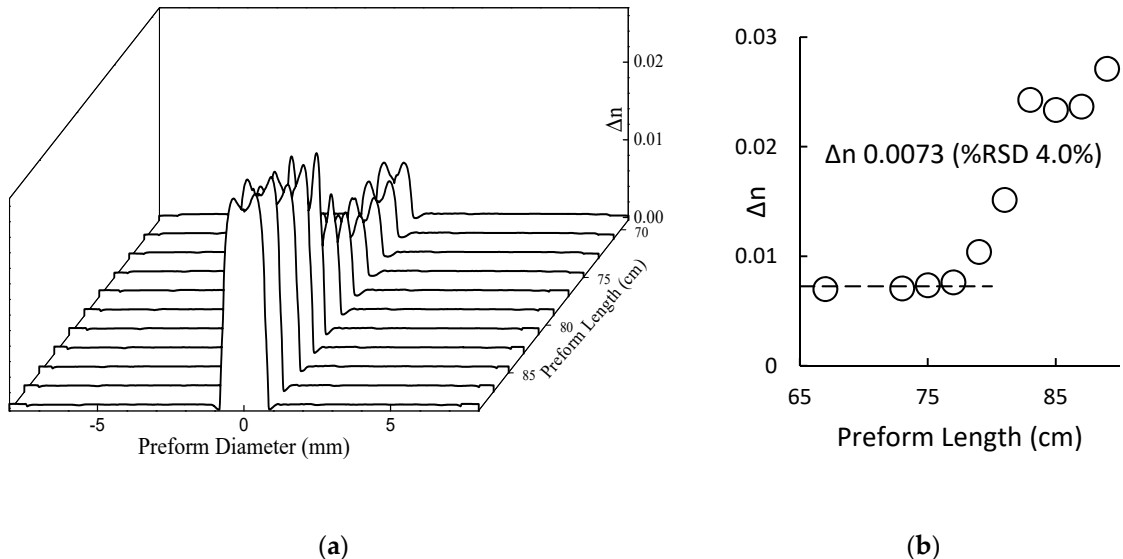

**(a)**                                                                      **(b)**

**Figure 7.** (**a**) The refractive index profile and (**b**) the longitudinal variation in the refractive index difference (Δn) for preform P3.

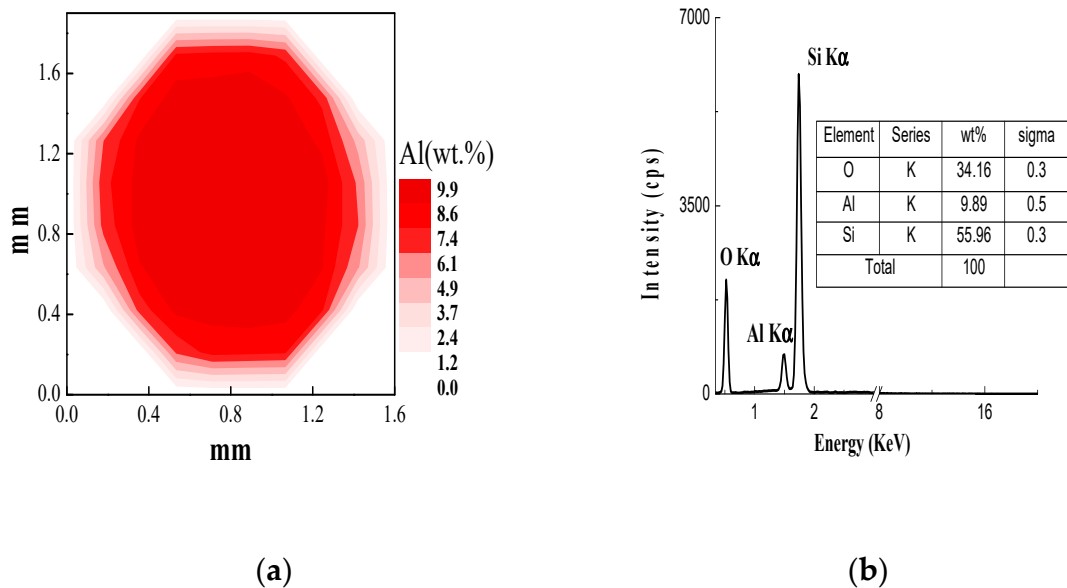

**(a)**                                                                      **(b)**

**Figure 8.** (**a**) A contour plot of the Al concentration in the core region of preform P3. The EDX point identification measurements were performed for 10 × 10 grid points covering an area of 2.0 × 2.0 mm. The sample for EDX analysis was cut from the edge of the preform adjacent to the outlet. (**b**) The EDX spectrum measured at the point with the highest Al concentration (9.9 wt.%). The inset table lists the wt.% of Al, Si, and O.

Figure 9a–c shows contour plots of the Al distribution for fibers F1, F2, and F3, respectively, that were drawn from preforms P1, P2, and P3, respectively. In general, the Al distributions for all fibers show the same pattern, with the highest Al content being located at the center of the core area. The maximum Al concentrations for fibers F1, F2, and F3 were 1.7, 4.3, and 3.2 wt.%, respectively. This corresponds to alumina concentrations of 1.9, 4.9, and 3.7 mole% for F1, F2, and F3, respectively. In the current study, the fabrication process efficiency was measured by dividing the alumina content obtained from EDX analyses of fibers by that derived theoretically. For the fabricated fibers F1 and F2, the process efficiency values were found to be 11 and 21%, respectively. These values indicate that the

$T_{Al}$ °C is a critical factor; an increase in $T_{Al}$ °C of 10 °C resulted in a 3.3-fold increase in $Al_2O_3$ content. The relationship between $\Delta n$ (obtained from RIP) and $Al_2O_3$ mol.% (obtained from EDX analysis) is plotted in Figure 10. As can be observed from the figure, the refractive index of silica increased by 0.0025 per mol% of $Al_2O_3$. This is in line with the results obtained by Bubnov et al. [25], where the authors used an MCVD vapor phase technique with $AlCl_3$ as the precursor.

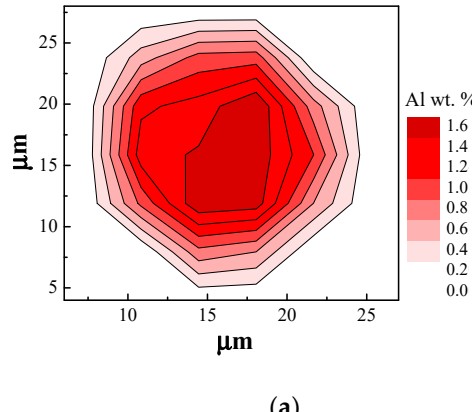

|     | wt.%  | σ wt.% | mole%     |      |
|-----|-------|--------|-----------|------|
| Si  | 45.28 | 0.4    | $SiO_2$   | 98.1 |
| O   | 53.06 | 0.3    |           |      |
| Al  | 1.66  | 0.09   | $Al_2O_3$ | 1.9  |

(**a**)

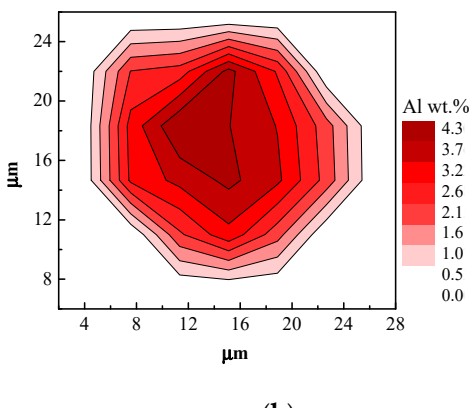

|     | wt.%  | σ wt.% | mole%     |      |
|-----|-------|--------|-----------|------|
| Si  | 42.95 | 0.5    | $SiO_2$   | 95.1 |
| O   | 52.75 | 0.3    |           |      |
| Al  | 4.30  | 0.1    | $Al_2O_3$ | 4.9  |

(**b**)

|     | wt.% | σ wt.% | mole%     |      |
|-----|------|--------|-----------|------|
| Si  | 43.9 | 0.5    | $SiO_2$   | 96.3 |
| O   | 52.8 | 0.4    |           |      |
| Al  | 3.22 | 0.1    | $Al_2O_3$ | 3.7  |

(**c**)

**Figure 9.** Contour plots of Al distribution across the core of fibers (**a**) F1, (**b**) F2, and (**c**) F3. The highest Al concentrations detected in F1, F2, and F3 were 1.7, 4.3, and 3.2 wt.%, respectively. The corresponding concentrations of $Al_2O_3$ were 1.9, 4.9, and 3.7 mole% for fibers F1, F2, and F3, respectively.

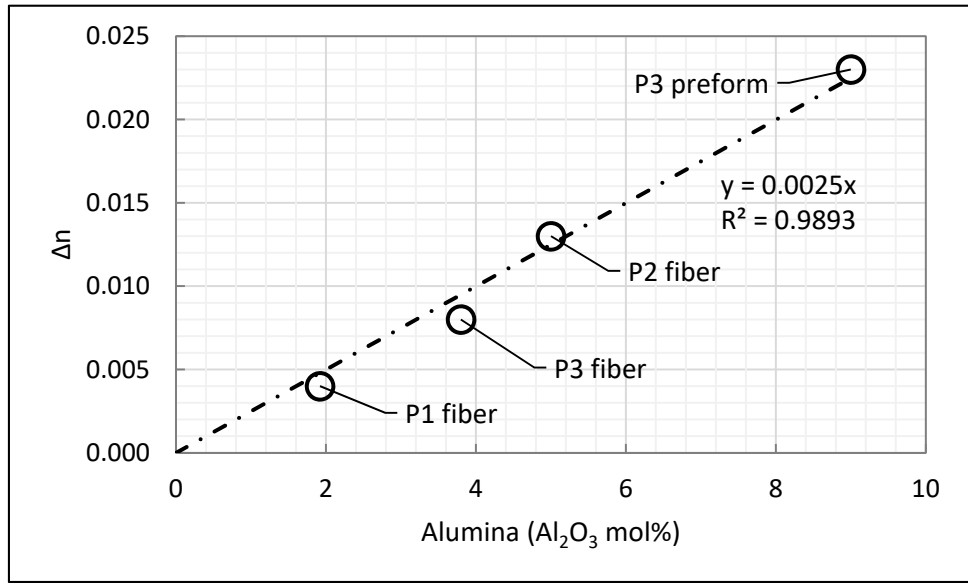

**Figure 10.** Plot of the refractive index difference vs. mol% of alumina in the range of 0 to 9 mol%. The refractive index of silica increases by 0.0025 per mol% of $Al_2O_3$.

## 4. Conclusions

Herein, we reported an initial study on alumina-doped silica preforms/fibers fabricated by an MCVD vapor phase technique using $Al(acac)_3$ as the precursor. The study, to the best of our knowledge, indicates the highest possible concentration of $Al_2O_3$ using this approach. The $Al(acac)_3$ sublimation temperature and the deposition direction (i.e., downstream and upstream) were varied while keeping other parameters, such as oxygen flow rate and deposition temperature, fixed. At a sublimation temperature of 185 °C, the process efficiency was found to be 21%. For the downstream deposition, the longitudinal uniformity for $\Delta n$ and core size was observed to be significantly higher than that for the combined downstream/upstream deposition (%RSD values for $\Delta n$ and core size of <2% and <2.4%, respectively, vs. 55% and 18%, respectively). We also reported on the refractive index change per mol% of $Al_2O_3$, which was found to be 0.0025. The optical and spectroscopic properties of the fabricated fibers are currently being thoroughly studied by our group.

**Author Contributions:** Conceptualization, K.A.M.S., N.Y.M.O., and H.A.A.-R.; methodology and investigation, K.A.M.S., N.Y.M.O., M.I.Z., and S.Z.M.Y.; data curation and analysis, K.A.M.S., M.I.Z., and S.Z.M.Y.; data validation, N.Y.M.O. and H.A.A.-R.; writing—original draft preparation, K.A.M.S.; writing—review and editing, N.Y.M.O., M.I.Z., S.Z.M.Y., and H.A.A.-R.; supervision, H.A.A.-R.; project administration, N.Y.M.O.; funding acquisition, H.A.A.-R. All authors have read and agreed to the published version of the manuscript.

**Funding:** The authors would like to thank Telekom Malaysia through its R & D subsidiary (TMR & D) for supporting this project.

**Conflicts of Interest:** The authors declare no conflict of interest.

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
