# Peer review of "Fabrication of Alumina-Doped Optical Fiber Preforms by an MCVD-Metal Chelate Doping Method"

_applsci, doi:10.3390/app10207231_

Round 1

Reviewer 1 Report

The manuscript entitled “Fabrication of Alumina Doped Optical Fiber Preforms by MCVD-metal Chelate Doping Method” has been carefully reviewed.

After review, I have some comments on the manuscript as below.

1) Was the alumina doped into the silica? Or was the alumina deposited on the inner surface of the silica tube? From the experimental method, it seems that the alumina was deposited on the inner surface of the silica tube rather than doped into the silica. The authors need more material analyses to clarify this question.

2) Al(acac)3 had been widely used as a CVD precursor for growing alumina. However, the authors did not give a detailed background description in the “Introduction” of the manuscript.

3) It is suggested that the “Materials and Methods” of the manuscript needs to be explained more clearly.

4) What kind of microstructure and morphology did the deposited alumina have? Particles or Films? Amorphous or crystalline?

5) The English writing of the manuscript needs extensive improvement.

Reviewer 2 Report

The paper is well written and interesting. However, my main concern is about the novelty of the presented results considering that the paper was submitted to Applied Sciences. Preforms as discussed in the paper are developed by commercial companies. Silica glass technology has been developed in the previous century. What is the novel contribution of this work? I think that the authors have to ask themselves very seriously this question and attempt to answer it in a more convincing way than it is currently the case in the manuscript.

Reviewer 3 Report

The presented paper reports on the fabrication of alumina-doped silica preforms via modified chemical vapor deposition. Authors use chelate delivery system with acetylacetonate ((Al)acac3)as a precursor. The effect of Al(acac)3 sublimator temperature and deposition direction on the fabrication process was investigated. Parameters such as refractive index and Al2O3 concentration were evaluated and compared.

This work has several significant shortcomings:

1) Formal shortcomings – the text is not logically structured. In Introduction, there is a description of the chelate delivery system used. I think this belongs more to the section Materials and Methods. Also a borderline between Materials and Methods and Results and Discussion is not clear – the section Results and Discussion begins with the description of Figure 3. I also lack some evaluation of the results and their impact.

2) The aim and motivation of the study is not clearly stated, as well as the impact of research on current applications is not mentioned.

3) In Introduction, line 29, is mentioned index raiser. Because it is the first mention of this parameter, it is better to use “refractive index” to better understanding.

4) There is no explanation what is %RSD mentioned (firstly mentioned in line 124) and how does it calculated?

5) Line 145 – “is manifested” should by placed with “is manifested”.

6) Graphic design of Table 1 is very strange and confusing. There is too much information and their connection is not clear.

Recommendation: Major revisions

Round 2

Reviewer 1 Report

The revised manuscript has been carefully reviewed. Unfortunately, I found that the quality of the article has not been improved and the authors did not make a good response to the first-review comments on the manuscript.

The revised manuscript still does not provide any analytical evidence to prove that the alumina was doped into the silica or the alumina was deposited on the inner surface of the silica tube. Moreover, the manuscript also lacks many necessary material analyses.

In addition, the authors’ reply to the comment 2 is as follows:

Comment 2) Al(acac)3 had been widely used as a CVD precursor for growing alumina. However, the authors did not give a detailed background description in the “Introduction” of the manuscript.

The authors reply: This is true for MOCVD. To the best of our knowledge, only one research group used Al(acac)3 as a precursor in an MCVD – Chelate delivery system setting. We included a reference to the study carried out by that research group [lines 48,49; reference [16]).

However, using Google Scholar to search with "Al(acac)3", "Al2O3" and "CVD" as keywords, you can find more than 2,000 related articles. The description in the Introduction of the manuscript seems that only a few studies use "Al(acac)3", but in fact it is not.

In general, in its current form, this article looks like a report on an undergraduate engineering experiment course, not a research paper.

Author Response

Thank you for the comment.

1. On the analytical evidence to prove that the alumina was doped into the silica or the alumina was deposited on the inner surface of the silica tube - we point to figure 8 (a) which shows a contour plot of Al concentration in the core region of preform P3. The EDX point identification measurements are performed for 10 × 10 grid points covering an area of 2.0 × 2.0 mm of the sample, cut of the edge of the preform adjacent to the outlet.

In figure 8 (b) EDX spectrum measured at the point with the highest Al concentration (9.9 wt.%) indicates the presence of Al, Si and O, shown in the inset table lists the wt.% of Al, Si and O.

We also point to figure 9 that shows the contour plots of Al distribution across the core of fibers (a) F1, (b) F2 and (c) F3. The highest Al concentration detected in F1, F2 and F3 is 1.7, 4.3 and 3.2 wt.%, respectively. Both figures 8 and 9 indicates that the Al was doped into the Silica in the core of the preform and fiber.

Further to this, we also argue the same by referring to figure 10 which plots the refractive index difference vs. mol% of alumina in the range of 0 to 9 mol%. The refractive index of silica increases by 0.0025 per mol% of Al2O3, which we argue to be evidence of Al being doped into the Silica in the core of the fiber.

2. On the use of Al(acac)3 as a CVD precursor for growing alumina, we have included citations to references listed in the introduction related to Al chelate and AlCl3 in the context of MCVD (Line 47 and below). We wrote:

"In most of these studies, anhydrous aluminum chloride (AlCl3) has been used as the precursor for alumina with few using aluminum acetylacetonate (Al(C5H7O2)3; Al(acac)3) as a precursor [e.g., 16]. AlCl3 is commonly used due to its relatively high vapor pressure.

Our search using the terms "Al(acac)3" and "MCVD" as keywords, returned only 30 articles, indicating only a few studies use "Al(acac)3". 

Reviewer 2 Report

This version of the paper is a large improvement on the previous one. It is now clear that the authors try to increase the content of Al2O3. Now the question to answer is: Does this contribution make any progress beyond the state of the art? The conclusions and introduction do not make any claims in this respect. In my view the authors should clarify if this paper makes any contribution beyond the state of the art or not. If the paper does not make any contribution then it should be rejected.

Author Response

Thank you for your comments.

On the matter of novelty, we report our latest findings related to achieving the highest possible doping level of Al2O3 using the chelate approach, in order to allow a higher concentration of rare earth elements in the Silica fiber. We wrote in the "Introduction":

"In this work, we studied the fabrication process of alumina doped silica fiber preforms with different Al(acac)3 sublimator temperature, (TAl°C) and deposition direction (i.e. downstream and upstream). Other parameters such as total oxygen flow, deposition temperature, carriage speed (Vb), and spindle rotation were fixed. The aim is to achieve the highest possible doping amount of alumina which in turn allows for higher incorporation of rare-earth elements into silica fibers. The fabricated preforms were checked for radial and longitudinal uniformity. The efficiency of the fabrication process was also determined by comparing the Al2O3 concentration obtained from energy dispersive X-ray spectroscopy (EDX) analyses with that derived theoretically. It is hopeful that the outcome of this study will add to the growing research concerning the MCVD-vapor phase fabrication of (highly) rare-earth doped silica fibers for high-power fiber lasers and fiber amplifiers."

Also in the "Conclusion", we humbly claim the novelty as such:

"We reported an initial study on alumina doped silica preforms/fibers fabricated by MCVD vapor phase technique using Al(acac)3 as the precursor. The study, to the best of our knowledge, indicates the highest possible concentration Al2O3 using this approach."

We believe that these statements in the 'Introduction' and 'Conclusion' indicates the novelty of the study being reported, while not to assert an overclaim.

Reviewer 3 Report

Dear Authors,

thank you for responding to the comments. Some parts were claryfied sufficiently (section Materials and Methods), but I still miss:
1) motivation and background for this research
2)I still lack some evaluation of the results and their impact
3) scientific overlap and improvement in the frame of potential application

I suggest accept the manuscript with major revisions.

Author Response

Thank you for your comments.

On the following matters:

1) motivation for this research

We wrote in the Introduction (line 60):

The aim is to achieve the highest possible doping amount of alumina which in turn allows for higher incorporation of rare-earth elements into silica fibers.

2) The background of the research, we wrote (from line 43 onwards):

Another alternative technique is the MCVD with chelate delivery system which was first reported by Tumminelli et al. [9]. This technique offers in-situ vapor phase deposition of dopants in a controllable process thus improving the quality of the fabricated preform or fiber. Since the study by Tumminelli et al., several research groups have reported on the use of MCVD with chelate delivery system [e.g., 10–16 17]. In most of these studies, anhydrous aluminum chloride (AlCl3) has been used as the precursor for alumina with few using aluminum acetylacetonate (Al(C5H7O2)3; Al(acac)3) as a precursor [e.g., 16]. AlCl3 is commonly used due to its relatively high vapor pressure.

3) On the evaluation of the results and their impact, we have discussed the relationship between fabrication process of alumina doped silica fiber preforms with their physical and material attributes . 

Among others, the fabricated preforms were checked for radial and longitudinal uniformity. The efficiency of the fabrication process was also determined by comparing the Al2O3 concentration obtained from energy dispersive X-ray spectroscopy (EDX) analyses with that derived theoretically. 

4) scientific overlap and improvement in the frame of potential application

We believe that the outcome of this study will add to the growing research concerning the MCVD-vapor phase fabrication of highly rare-earth doped silica fibers for high-power fiber lasers and fiber amplifiers. 

Round 3

Reviewer 1 Report

In the reviewer opinion, the manuscript does not report a new study in the field and the authors have not well answered to the previous questions. Moreover, although the manuscript has been revised twice, the authors did not add any additional experimental evidence to the manuscript and the quality of the manuscript has not been improved significantly. Therefore, in the reviewer opinion, the manuscript cannot be published in the current form.

Reviewer 2 Report

I am satisfied with changes.

Reviewer 3 Report

Dear Authors,

thank you for your reaction. I think that you improved the paper enough to be published.